# Influence of Fracture Reduction on the Functional Outcome after Intramedullary Nail Osteosynthesis in Proximal Humerus Fractures

**DOI:** 10.3390/jcm11226861

**Published:** 2022-11-21

**Authors:** Annika Hättich, Tim Jodokus Harloff, Hatice Sari, Carsten Schlickewei, Christopher Cramer, André Strahl, Karl-Heinz Frosch, Konrad Mader, Till Orla Klatte

**Affiliations:** Department of Trauma and Orthopaedic Surgery, University Hospital Hamburg Eppendorf, Martinistraße 52, 20246 Hamburg, Germany

**Keywords:** shoulder, fracture reduction, proximal humerus

## Abstract

Background: Optimal treatment of proximal humeral fractures (PHF) is still controversial. Therefore, we aim to investigate the influence of fracture reduction after intramedullary nailing of a PHF on the clinical outcome. Methods: Patients treated with intramedullary nail for PHF from 2013–2018, (minimum follow-up 12 months) were retrospectively included. Constant Score (CS), DASH and Simple Shoulder Test (SST) were collected. Postoperative radiographs were used to determine head-shaft-alignment (HSA), head-shaft-displacement (HSD), integrity of the medial hinge and the cranialization of the greater tuberosity (CGT). The results of fracture reduction were categorized as either “anatomical”, “acceptable” or as “malreduction”. Malreduction exists when at least one of the following parameters are present: HSA > 150/< 110°, HSD >5 mm, CTM > 5 mm or lack of integrity of the medial hinge. Results: 42 patients (mean age 65.5 ± 14.1 years, 15 male, 27 female) with a mean follow-up time of 43 months were included. The average CS was 60 ± 30, DASH 49.8 ± 24.3 and SST 62.9 ± 26.9. There was an “anatomic” reduction in 9 (21.4%), “acceptable” in 7 (16.6%) and a “malreduction” in 26 (62%) patients. Comparing the combined “anatomical” and “acceptable” reduction group with the “malreduction” group, worse scores were observed in the “malreduction” group (CS 67.2 vs. 55.2, DASH 45.2 vs. 51.9, SST: 69.3 vs. 58.6) without statistical significance (CS: *p* = 0.095, DASH: *p* = 0.307, SST: *p* = 0.400). By means of multiple logistic regression analyses no statistically significant risk factors were identified for lower DASH, CS and SST scores. Conclusions: Anatomical fracture reduction using intramedullary nails has a positive effect on postoperative outcome for the 3 scores recorded, without reaching statistical significance.

## 1. Introduction

Fractures of the proximal humerus are among the most common fractures, especially in the older generation [1]. To this date, the treatment of proximal humerus fractures remains controversial [2,3,4]. Therapeutic options range from conservative therapy to various osteosynthesis techniques and prosthetic treatment. Although several studies have already demonstrated that the benefit of surgical therapy outweigh those of conservative treatments for dislocated fractures, the best treatment remains unclear.

Furthermore, several studies have already identified factors that positively influence the clinical outcome [5,6,7,8,9]. They used biomechanical and histologic studies to demonstrate which fracture types are at particularly high risk for developing humeral head necrosis. Hertel et al. [8] did establish radiological criteria that could affect the blood circulation of the humeral head and therefor the postoperative healing. Schnetzke et al. [9] transferred these data to the treatment of humeral head fractures by using locking plates. He concluded that anatomical reduction and stabilization using locking plates led to a significantly improved outcome compared to insufficient anatomical reduction according to Hertel’s criteria. 

In analogy to this work, we examined the functional outcome after using an intramedullary nail as treatment of proximal humerus fractures. We also investigated the functional outcome in relation to the radiological results of fracture reduction. 

## 2. Methods 

### 2.1. Study Population

This retrospective study included all patients who had been treated in our hospital (Level 1 Trauma Center) for proximal humerus fractures with an intramedullary nail during 2013–2018. The presence of a postoperative radiograph in two levels (a.p. and y-view) was also mandatory. Exclusion criteria were preexisting diseases that affect the mobility or strength in the affected shoulder joint, previous operations of the affected shoulder, the implementation of an immunosuppressive therapy, lack of patient consent, lack of compliance, or legal incompetence. 

The minimum follow-up time was set at 12 months. Vulnerable groups (underaged persons, pregnant women, non-consenting individuals) were excluded as well as patients with repeated revision surgery, e.g., after infection, postoperative hematoma or subsequent arthrolysis. The study was approved by the ethics committee of Hamburg Medical Association (PV6086).

### 2.2. Surgical Technique

All patients were treated with a mini-open reduction followed by an intramedullary nail osteosynthesis (Trigen Meta Nail, Smith & Nephew, Inc. 7135 Goodlett Farms Parkway Cordova, TN, USA). 

Additionally, in case of tuberosity involvement, a refixation with an additional suture cerclage was performed. The nail was implanted via skin incision of approximately 5 cm in length as a lateral deltoid split, starting from the lateral acromion angle. After incision of the fascia and rotator cuff, the nail was inserted at the most cranial point of the humeral head after reduction of the fracture. Proximal and distal locking was performed using a targeting device under fluoroscopic surveillance. 

### 2.3. Classification

Fracture classification according to AO and Neer [10] was performed by two blinded observers at 2 time points to evaluate the intra- observer reliability of the classification using the available preoperative radiographs in anterior-posterior and y-view in addition to the inter- observer reliability.

### 2.4. Radiographic Evaluation

Radiographic evaluation of the postoperative images was performed by one blinded investigator. All images were obtained during the second day after surgery. 

Radiographs in anterior-posterior view as well as y-view were available in each case. 

In this blinded analysis, the following criteria were evaluated based on the study by Hertel et al. [8] mentioned above. The head-shaft-alignment, head-shaft-displacement, the integrity of the medial hinge and cranialization of the greater tuberosity (CGT) were measured (described in Figure 1).

Depending on the parameters displayed in Table 1, the fracture reduction was classified as “anatomical”, “acceptable” or “malreduction”. 

An “anatomical” result was present if all 3 anatomical parameters were achieved. 

For the “acceptable” group, at least 1–3 parameters of the “acceptable” category had to be measured but no parameter of the category “malreduction”. As soon only one parameter of the “malreduction” category was observed, the patient was classified into this group.

### 2.5. Clinical Examination and Scores

All patients were examined by two investigators who were blinded for the postoperative radiographs and fracture’s classification at the time of the examination. The clinical examination included an evaluation of range of motion for anteversion, abduction and external rotation. Furthermore, patients were evaluated using the Constant Score (CS) [11,12]. During the examination, a standardized force measurement was performed for abduction on the 30° anteverted arm. In addition, we used the Disabilities of Arm, Shoulder and Hand (DASH) score and the Simple Shoulder Test (SST) to subjectively assess the shoulder function [13]. 

In correlation to the radiographic evaluation, we evaluated if the quality of fracture reduction had a significant effect on the postoperative clinical outcome and the results of the collected score values. For this purpose, the anatomical and acceptable groups were combined to “group 1” and compared to the malreduction group “group 2”.

### 2.6. Statistical Analysis

Continuous variables were expressed as mean ± standard deviation (SD), while categorical variables are expressed as a number and percentage (%). Normal distribution analysis was performed using the Kolmogorov–Smirnov-test and the Shapiro–Wilk test. In case of normal distribution, t-test for independent variables, and Mann–Whitney U test for non-normal distributed variables were applied to test whether there was significant difference between two defined groups related to the clinical outcome. Further, Fisher exact test was used to investigate whether two dichotomous characteristics were independent. As this study describes a retrospective data analysis with a small number of cases, a patient matching was not possible. Therefore, the results of the group comparison might not have enough statistical power to detect significant differences. To counteract this fact, the basic statistics were supplemented with a multiple logistic regression analysis using all cases of the dataset for analysing potential risk factors for malreduction as well as low DASH, CS and SST result (with their corresponding odds ratios (OR).

Additionally, the inter-observer reliability was determined by calculating the kappa values based on a multi-rater kappa-equation. Multi-rater kappa summarizes the strength of agreement for all possible comparisons between the observers including the same and mixed types of experience levels [14]. The kappa statistics was used as the chance-corrected measurement of agreement and was interpreted as being perfect in the range of 0.81 to 1.00, good in the range of 0.61 to 0.80, moderate in the range of 0.41 to 0.60, fair in the range of 0.21 to 0.40, and poor in the range of 0.00 to 0.29 according to the definitions of Landis and Koch [15]. The level of significance for all tests was set at *p* < 0.05. All data were analyzed using IBM SPSS Statistics version 29.0 (IBM, Armonk, NY, USA).

## 3. Results

### 3.1. Demographics

In total, we included 42 patients with a mean follow-up time of 43 ± 22.5 months (min.: 12.6, max.: 88.5). Further detailed demographic information are listed in Table 2.

### 3.2. Classification

Among the 42 cases, 5 fractures according to AO classification 11A were present (1 × 11A2, 4 × 11A3, 12%). In 19 cases we observed the fracture type 11B1 (45.2%). C fractures were present in a total of 42.9%, including 7 × 11C1 (16.7%), 5 × 11C2 (12%), and 5 × 11C3 (12%) (Figure 2). 

13 patients were classified as Neer III (31%) and 29 as Neer IV (69%) based on the Neer classification. The level of agreement of inter-observer reliability between observer 1 and 2 was rated as “good” for the AO classification and “fair” for the Neer classification (Table 3).

### 3.3. Radiographic Evaluation

Based on the defined criteria, we found an anatomical reduction in 21.4%, an acceptable reduction in 16.7% and a malreduction in 61.9%. A more detailed analysis of the individual parameters of fracture reduction divides the collective for head-shaft displacement into 31% “anatomical”, 16.7% “acceptable” and 52.4% “malreduction”. In the evaluation of the CGT, mainly anatomical reduction results were seen (“anatomical”: 83.3%, “acceptable” 2.4%, “malreduction” 14.3%). For head-shaft-alignment, “anatomical” results were seen in 66.6%, “acceptable” in 2.4%, and “malreduction” was present in 31%. 88.1% of the cases had an anatomic integrity of the medial hinge, 11.9% had non-anatomic results (5 cases without integrity of the medial hinge, Figure 3). 

### 3.4. Clinical Examination and Scores

In the evaluation of CS, both the operated and the healthy shoulder were examined. On the operated side, we were able to determine an overall averaged Constant Score of 60 points (60.0 ± 23.0, min.: 16, max.: 99). In contrast, on the healthy shoulder, we observed an average CS of 82.2 points (±9.5, min.: 55, max.: 100). There is a significant difference between the operated and the healthy shoulder (*p* < 0.001). The mean DASH was 49.8 (±24.3), the SST 62.9 (±26.9) (Table 4).

Based on the classification according to the results of the radiological parameters, the score results were compared between the groups “anatomical/acceptable” (group 1) vs. malreduction (group 2). The mean CS of Group 1 was 67.2 (±22.4, min.: 16, max.: 99) and the result of group 2 was 55.2 (±22.5, min.: 19, max.: 88). Even if there was an observed difference of at least 10 points between the two groups with worse results for group 2, no statistically significant difference was observed (*p* = 0.095). Additionally, non-significant differences were found in DASH Score between the two groups. The DASH in group 1 was 45.2 (±24.4, min.: 24.1, max.: 100), in group 2 51.9 (±21.9, min. 24.1, max. 91.4), indicating a better result for group 1, but no statistical significance was observed (*p* = 0.307). Additionally, the SST showed with an average value of 69.3 (±30.1, min.0, max. 100) for group 1 and 58.6 (±24.2, min.: 8.3, max.: 91.7) for group 2 no significant differences (*p* = 0.400) (Figure 4, Table 5).

### 3.5. Risk Factor Analysis

A total of 42 cases were included in a multiple regression analysis model that investigated independent risk factor for malreduction and the functional outcome scores. The analysis used the “Enter” method to examine the significant impact of all variables simultaneously. The regression model for malreduction was statistically significant (χ^2^ = 44.98, *p* < 0.001) and identified one main independent predictor associated with malreduction, as shown in Table 6. The patients age and gender had a priori no influence on the radiological outcome. 

The model explained 89.4% (Nagelkerke R^2^) of the variance and correctly classified 95.2% of cases. The results indicate that especially in case of an increasing head-shaft-displacement, the odds of malreduction significantly increased. 

In correspondence with our outcome scores comparison between group 1 and 2, further regression analyses were applied to investigate the influence on the functional outcome operationalized by the DASH, CS and SST Score each stratified by patients with <50 and ≥50 points. The analysis identified no single significant predictor for DASH (χ^2^ = 6.207, *p* = 0.184), SST (χ^2^ = 8.967, *p* = 0.054) or CS (χ^2^ = 8.967, *p* = 0.062). Neither age, gender, malreduction or preoperative AO classification had a significant influence on the clinical outcome. All investigated singe variables demonstrated a *p*-value > 0.05 in all performed analyses.

## 4. Discussion

Treatment options for dislocated proximal humerus fractures remain controversial. While there is consensus that undislocated fractures should be treated conservatively, there remains disagreement regarding the treatment of rather complex fracture types [2,3,6]. In this study, the outcome of fracture reduction using an intramedullary nail was investigated. 

Schnetzke and colleagues [9] were already able to demonstrate a positive benefit of anatomical fracture reduction on the postoperative outcome of fracture treatment using an angle-stable plate. Despite satisfactory results overall, he also found a high rate of inadequate reduction of almost 60%. Compared to our study, he investigated only fracture type C according to AO-classification. In total, he investigated 98 patients (mean age 61.1 years) with a follow up rate of 3 years using the same radiographic measurements and clinical scores. He found an anatomical or acceptable fracture reduction (group 1) in 40.8% of his collective, while 59.2% had a “malreduction”. Like in our investigation, he stated a significant better clinical outcome and a lower complication rate in the anatomical group 1 compared to group 2. 

Still, the results of our study are quite similar. Overall, only 21.4% of our patients achieved an “anatomic” fracture reduction according to the criteria above, whereas 16.6% were still “acceptable” and even 62% fell into the “malreduction” group. This suggests that nail osteosynthesis often fails to achieve an anatomical reduction in complex fracture types. This conclusion was also seen by Blum et al. [16], who demonstrated in a retrospective follow-up study the worst results in CS and DASH in type C fractures after nail osteosynthesis. Compared to our recent study, he only investigated the clinical results, but did not consider the fracture reduction. 

Fortunately, in our study, anatomic positioning of the medial hinge was seen in 88.1% of the cases, which is an improvement compared to other studies, especially also to the ones that use angular stable plates [9,17]. However, poor reduction can be seen while using nailing in 52% for head-shaft displacement in our collective. This corresponds to the results obtained with plate osteosynthesis (51%). Considering the other criteria, the intramedullary nail achieved a significantly poorer value of 31% malreduction for the “alignment” criterion compared to plate osteosynthesis. 

The mean absolute Constant score of our collective was 60 points ± 23.0, which is a better value compared to the work of Schnetzke et al. [9] (54.8 points). We must complement that our collective included more patients with a fracture of AO classification type B. Compared to the patients’ healthy shoulder, where we determine a mean CS of 82.2 points, there is a significant difference in shoulder function (*p* < 0.001) between both sides. These results insists that the postoperative shoulder function is significantly worse 43 months after surgery. The same results were seen at Blum’s study [16].

Although the CS of the operated shoulder does not show a statistically significant difference between the two groups, there is nevertheless a striking difference in the absolute score of 67.2 points in group 1 to only 55.2 points in group 2. Even though the use of clinical scores is critically discussed, especially in view of smaller case numbers [18], we consider the difference of 12 points between the two groups as a clear indication that fracture reduction is a determining factor for the postoperative outcome when using the intramedullary nail. 

The same result is reflected in the DASH score and SST. 

Our clinical results are in the range of the reference values of the current literature, concerning both studies using nail and plate osteosyntheses. Even if it was not the main aim to investigate the benefits of nailing compared to angle stable plating, we wanted to compare our results to the current status in the literature. Shi et al. performed a systematic review and meta-analysis examining the postoperative outcomes after using the two mentioned procedures [19]. Here, advantages were seen for nailing in terms of complication rate, intraoperative blood loss, operative time, and humeral head necrosis—however, there was also a need for further and larger studies in terms of fracture type and indication.

In contrast, Gracitelli et al. conducted a prospective randomized control trial that demonstrated the opposite [20]. Again, little difference was seen in clinical outcome and score results between nailing and plaiting, but there was a significantly higher complication rate in the group that had been treated by nail. 

Boudard et al. [21] as well as von Rüden et al. [22] compared the two surgical techniques. Both found no significant difference between the two methods. However, Boudard and colleagues noted that the current trend is more towards plating for fracture treatment, especially for 3–4-part fractures. 

Another point to discuss referring to proximal humerus fractures is the classification to be used. Currently, mainly the classifications according to AO, Neer or Codman are common. However, poor inter-observer reliability has been found in the literature, especially for the Neer, but also for the AO classification [23]. In our study, we were able to determine a “good” reliability for the AO classification and a “fair” reliability for the Neer classification. 

This study has some clear limitations and assumptions. The major limitation of this study, beside its retrospective design, was the small number of patients due to a high rate of lost to follow up. This is certainly also due to the high patient age of the collective. Several attempts were made to contact these patients but no actual information was possible to obtain from the individuals or their families. Another limitation of the current study is the definition of fracture reduction and if this bad, acceptable or good. We choose this classification according to the study of Schnetzke et al. [9]. Reason for this was that we tried to achieve a best possible comparability between their (plate ORIF) and our treatment (nail CRIF) because we could not conduct a case–control study based on our study population.

## 5. Conclusions

Treatment options for dislocated proximal humerus fractures remain variable. Despite the lack of statistical analysis anatomic fracture reduction has a decisive influence on the postoperative clinical outcome after intramedullary nail osteosynthesis.

## Figures and Tables

**Figure 1 jcm-11-06861-f001:**
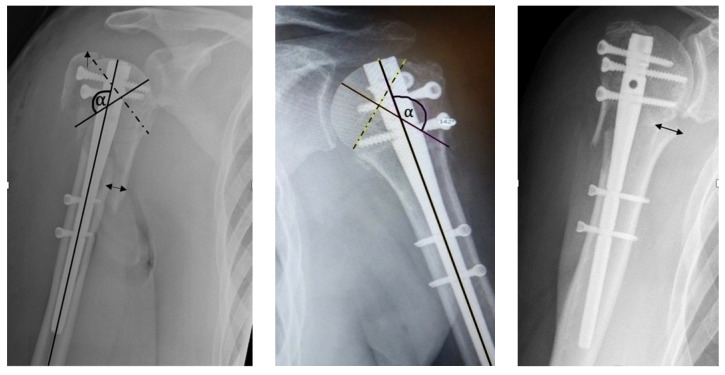
Left: Measurement: (1). Head-Shaft-Alignment (angle Alpha): We drew a first line (dashed) from the superior border to the inferior border of the articular surface. A second line was drawn perpendicular to the first line through the center of the humeral head. The third line shows the alignment of the humeral shaft. The angle alpha between the second and third line describes the head shaft alignment. (2). Head-Shaft-Displacement (two-way-arrow): Measurement between the medial shaft fracture line and the medial edge of the head (mm). (3). CTB (one-way-arrow): Measurement between the greater tuberosity and the cartilaginous fracture line (mm). [9]. Middle: Anatomical Reduction, using the described criteria Right: Head-Shaft-Displacement 15.4 mm.

**Figure 2 jcm-11-06861-f002:**
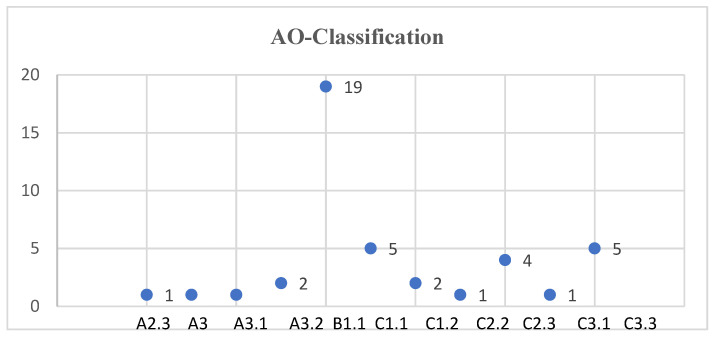
Distribution of fracture type according to AO-Classification.

**Figure 3 jcm-11-06861-f003:**
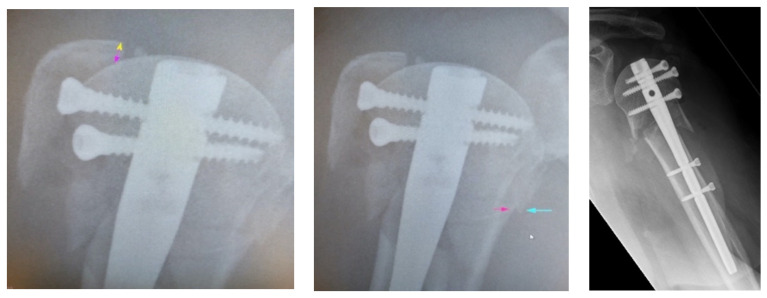
Left: Cranialization of GTB, Middle: Missing integrity of the medial hinge. Right: Head-Shaft-Alignment 196°.

**Figure 4 jcm-11-06861-f004:**
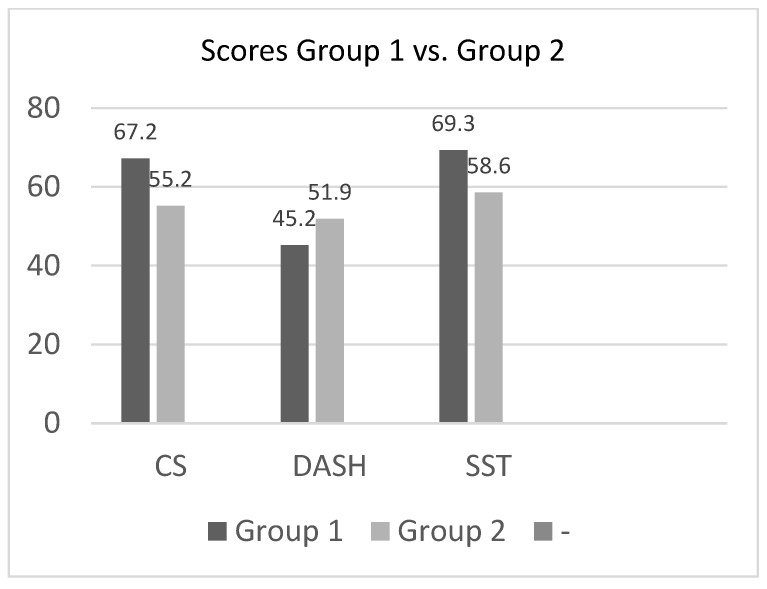
Scores comparing group 1 and group 2.

**Table 1 jcm-11-06861-t001:** Parameters of fracture reduction.

Anatomical Parameter	Anatomical	Acceptable	Malreduction
Head-Shaft-Displacement	Anatomical	<5 mm	>5 mm
Head-Shaft-Alignment	Normal, 120–150°	Minor varus, <120–100°	Valgus, >150°, <110°
Cranialization of the Greater Tuberosity	Anatomical	<5 mm	>5 mm

**Table 2 jcm-11-06861-t002:** Demographics, radiographic assessment, and functional outcome in 42 patients with humerus fracture.

Parameter	N (%) or Mean ± SD (Range)	Data Distribution (Normality Test)
Gender male female	15 (35.7%) 27 (64.3%)	---
Age [years]	65.5 ± 14.4 (22.9 to 86.1)	failed *p* < 0.001 ^†^ *p* < 0.001 ^‡^
Handedness right left	38 (90.5%) 4 (9.5%)	---
Injured side right left	24 (57.1%) 18 (42.9%)	---
Injured side dominant non-dominant	27 (64.3%) 15 (35.7%)	---
Neer-classification III IV	13 (31.0%) 29 (69.0%)	---
Follow-up [months]	43.0 ± 22.5 (12.6 to 88.5)	failed *p* < 0.001 ^†^ *p* < 0.001 ^‡^

† Kolmogorov–Smirnov test, ‡ Shapiro–Wilk test.

**Table 3 jcm-11-06861-t003:** Inter-observer reliability of radiographic assessment of humerus fracture classification between observer I (fellowship trained trauma surgeon) and observer II (clinical traumatology fellow).

	Kappa Values	*p* Value	Percentage Agreement	Level of Agreement ^a^
AO classification	0.783	<001	35/42 (83.3%)	good
Neer classification	0.380	0.002	33/42 (78.6%)	fair

^a^ based on the definitions described by Landis and Koch [15]. AO, Arbeitsgemeinschaft für Osteosynthesefragen (working group for bone fusion issues).

**Table 4 jcm-11-06861-t004:** Results of the clinical scores in total.

DASH score	49.8 ± 24.3 (24.1 to 121.6)	Failed *p* = 0.022 ^†^, *p* = 0.001 ^‡^
SST score	62.9 ± 26.9 (.0 to 100.0)	Passed *p* = 0.107 ^†^, *p* = 0.074 ^‡^
Constant Score injured side	60.0 ± 23.0(16 to 99)	Failed, *p* = 0.006 ^†^, *p* = 0.004 ^‡^
Constant Score healthy side	82.2 ± 9.5 (55 to 100)	Passed *p* = 0.200 ^†^, *p* = 0.036 ^‡^

‡ using Mann–Whitney U test, † Fisher’s test. DASH: Disabilities of Arm, Shoulder and Hand, SST: Simple Shoulder Test.

**Table 5 jcm-11-06861-t005:** Scores comparing group 1 (anatomical/acceptable fracture reduction) and group 2 (malreduction).

Score	Group 1	Group 2	*p*-Value
DASH score	45.2 ± 24.4 (24.1 to 100.0)	51.9 ± 21.9 (24.1 to 91.4)	*p* = 0.307
SST score	69.3 ± 30.1 (.0 to 100.0)	58.6 ± 24.2 (8.3 to 91.7)	*p* = 0.400
CS injured side	67.2 ± 22.4 (16 to 99)	55.2 ± 22.5 (19 to 88)	*p* = 0.095

**Table 6 jcm-11-06861-t006:** Logistic regression for independent predictors of malreduction in patients with proximal humeral fractures. Regression with “Enter”-method.

Regression Parameter							95% CI	
Predictor	Beta	SE Beta	Wald’s χ^2^	df	*p*-Value	OR	Lower	Upper
Age	0.353	0.195	3.259	1	0.071	1.42	0.97	2.08
Gender	−1.198	2.463	0.237	1	0.627	0.30	0.00	37.67
AO classification	−3.471	2.382	2.124	1	0.145	0.03	0.00	3.31
Head-shaft-displacement	1.917	0.888	4.185	1	0.041	6.15	1.08	35.08
Head-shaft-alignment	0.181	0.135	1.998	1	0.158	1.21	0.93	1.58

Overall model evaluation Omnibus test Chi^2^ = 44.98, *p* < 0.001; Goodness-of-fit test Hosmer–Lemeshow test Chi^2^ = 4.131, *p* = 0.845.

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
