# Peer review of "Influence of Fracture Reduction on the Functional Outcome after Intramedullary Nail Osteosynthesis in Proximal Humerus Fractures"

_jcm, 2022, doi:10.3390/jcm11226861_

Round 1

Reviewer 1 Report

The reviewer has several questions for the authors before considering the MS for publication.

1. Fig. 2

The right panel seemed to have a compromised medial hinge, which is known to be a negative factor impacting fixation quality and post-op patient outcome. Why did not the surgeons correct it during the operation? How many cases were with compromised medial hinge? 

2. Table 6.

1) The scoring was not statistically significant. Therefore, the reviewer does not believe any conclusion could be drawn from the data. 

2) It is believed that the study should have been a case-control study investigating the risk factors for implant failure and bad patient outcomes. Quality of fracture reduction, fracture type, involvement of medial hinge, etc, should be the potential risk factors that would be analyzed by logistic regression or any other statistical methods where appreciated.  There is no way for the reviewer to agree with the scientific merit of the current study design, results, or conclusion.

3. The Conclusion is not supported by the data, which apparently did not show any statistical significance.

Author Response

Reviewer #1

  1. Fig. 2

The right panel seemed to have a compromised medial hinge, which is known to be a negative factor impacting fixation quality and post-op patient outcome. Why did not the surgeons correct it during the operation? How many cases were with compromised medial hinge? 

We agree with the reviewer. Since this is a retrospective study, we cannot provide a rationale for leaving the malreduction in place. The malreposition would no longer be accepted in this way today in our clinic.

There were 5 cases with compromised medial hinge (Line 178/179).

  1. Table 6.

1) The scoring was not statistically significant. Therefore, the reviewer does not believe any conclusion could be drawn from the data. 

We agree with the reviewer. At best, a trend can be assumed on the basis of the retrospective study population. We therefore corrected our conclusion.

2) It is believed that the study should have been a case-control study investigating the risk factors for implant failure and bad patient outcomes. Quality of fracture reduction, fracture type, involvement of medial hinge, etc, should be the potential risk factors that would be analyzed by logistic regression or any other statistical methods where appreciated.  There is no way for the reviewer to agree with the scientific merit of the current study design, results, or conclusion.

We apologize for the incorrect description of the statistical method used. Used tests were added in the statistical analysis section and as well in the table (tab 2, 5 and 6) when the test was used. The test for normal distribution was performed using the Kolmogorov-Smirnov test and the Shapiro-Wilk test.  The Mann-Whitney U test was used to test whether there is a difference between two groups. The Fisher exact test was used to test whether two dichotomous characteristics were independent.

  1. The Conclusion is not supported by the data, which apparently did not show any statistical significance.

We agree with the reviewer. Please see comment 2.1.

Reviewer 2 Report

This study evaluates the results of proximal humerus synthesis with intramedullary nail according to the degree of reduction.

The study question is very important; reduction is often a factor that is underestimated in intramedullary osteosynthesis. Such a study reiterating the importance of epiphyseal reduction is likely to make a positive contribution to the literature.

The study is simple, linear and clear, as well as methodologically well conducted. The article is generally well written: the introduction is brief but complete, the methods are reproducible, the results are clear, and the discussion informative. However, the conclusions are not consistent with the results in my opinion: statistics cannot be ignored. Certainly the functional scores were better in the group with good reduction, but without statistical significance, this should be emphasized in the conclusions.

Therefore, I think the article is worthy of publication, but I recommend some major revisions:

1) in the description of statistical methods, the type of univariate analyses performed should also be included

2) the iconography is poor, it would be helpful to include representative figures of each reduction criterion used

3) in the methods please better clarify the exclusion criteria

4) include a large paragraph in discussion to describe the limitations of this study

5) modify the conclusion to reiterate that this study did not show statistically significant differences, but that a generally better functional outcome resulted in the group with good reduction.

Thank you.

Author Response

Reviewer #2

1) In the description of statistical methods, the type of univariate analyses performed should also be included

We apologize for the incorrect description of the statistical method used. See answer reviewer #1 comment 2.2

2) The iconography is poor, it would be helpful to include representative figures of each reduction criterion used

Thank you for your comment. We added more pictures in Fig. 1 and Fig. 2 so each criterion can be seen clearly.

3) In the methods please better clarify the exclusion criteria

 Thank you for your comment. We added the exclusion criteria. Section Study Population line 60-70.

4) Include a large paragraph in discussion to describe the limitations of this study

We agree with the reviewer and added at the end of the discussion this paragraph (Line 301-309): This study has some clear limitations and assumptions. The major limitation of this study, beside its retrospective design, was the small number of patients due to a high rate of lost to follow up. This is certainly also due to the high patient age of the collective. Several attempts were made to contact these patients but no actual information was possible to obtain from the individuals or their families. Another limitation of the current study is the definition of fracture reduction and if this bad, acceptable or good. We choose this classification according to the study of Schnetzke at al.. Reason for this was that we tried to achieve a best possible comparability between their (plate ORIF) and our treatment (nail CRIF). Because we could not conduct a case-control study based on our study population.

5) Modify the conclusion to reiterate that this study did not show statistically significant differences, but that a generally better functional outcome resulted in the group with good reduction.

We agree and rephrased the conclusion.

Round 2

Reviewer 1 Report

The nature of the design is a fatal flaw. It is neither RCT nor case-control. None of the statistical methods in the paper were correct.  Multiple regression should be used for analyzing potential risk factors and OR be reported in the results, rather than simply comparing between two groups. It is highly suggested to consult a statistician to re-design the study and re-analyze the data before re-submission.

Author Response

"The nature of the design is a fatal flaw. It is neither RCT nor case-control. None of the statistical methods in the paper were correct.  Multiple regression should be used for analyzing potential risk factors and OR be reported in the results, rather than simply comparing between two groups. It is highly suggested to consult a statistician to re-design the study and re-analyze the data before re-submission."

 Thanks for your comment – we agree with your opinion so we supplemented the basic statistic with a multiple logistic regression analysis using all cases of the dataset for analysing potential risk factors for malreduction as well as low DASH, CS and SST result (line 126-132).

The results are described in line 217-238 as well as in table 7. Neither age, gender, malreduction or preoperative AO classification had a significant influence on the clinical outcome. All investigated singe variables demonstrated a p-value > 0.05 in all performed analyses.

Reviewer 2 Report

Authors have addressed all my concerns. Thank you.

Author Response

Thank you for your comment and the review. We did check the language again and made some final corrections